# AI-Enhanced Fluorescein Angiography Detection of Diabetes-Induced Silent Retinal Capillary Dropout and RNA-Seq Identification of Pre-Symptomatic Biomarkers

**DOI:** 10.3390/biomedicines13081926

**Published:** 2025-08-07

**Authors:** Yiyan Peng, Huishi Toh, Dennis Clegg, Peng Jiang

**Affiliations:** 1Center for Gene Regulation in Health and Disease, Cleveland State University, Cleveland, OH 44115, USA; 2Department of Biological, Geological and Environmental Sciences, Cleveland State University, Cleveland, OH 44115, USA; 3Department of Integrative Genomics and Epidemiology, Meharry Medical College, Nashville, TN 37208, USA; 4Neuroscience Research Institute, University of California Santa Barbara, Santa Barbara, CA 93105, USA

**Keywords:** image-AI, Bayesian framework, retinal capillary dropout prediction, RNA-seq, biomarkers

## Abstract

**Objective**: Retinal capillary dropout, characterized by acellular capillaries or “ghost vessels,” is an early pathological sign of diabetic retinopathy (DR) that remains undetectable through standard clinical imaging techniques until visible morphological changes, such as microaneurysms or hemorrhages, occur. This study aims to develop a non-destructive artificial intelligence (AI)-based method using fluorescein angiography (FA) images to detect early-stage, silent retinal capillary dropout. **Methods:** We utilized 94 FA images and corresponding destructive retinal capillary density measurements obtained through retinal trypsin digestion from 51 Nile rats. Early capillary dropout was defined as having an acellular capillary density of ≥18 counts per mm^2^. A DenseNet based deep learning model was trained to classify images into early capillary dropout or normal. A Bayesian framework incorporating diabetes duration was used to enhance model predictions. RNA sequencing was conducted on retinal vasculature to identify molecular markers associated with capillary early dropout. **Results**: The AI-based FA imaging model demonstrated an accuracy of 80.85%, sensitivity of 84.21%, specificity of 75.68%, and an AUC of 0.86. Integration of diabetes duration into a Bayesian predictive framework further improved the model’s performance (AUC = 0.90). Transcriptomic analysis identified 43 genes significantly upregulated in retinal tissues preceding capillary dropout. Notably, inflammatory markers such as Bcl2a1, Birc5, and Il20rb were among these genes, indicating that inflammation might play a critical role in early DR pathogenesis. **Conclusions**: This study demonstrates that AI-enhanced FA imaging can predict silent retinal capillary dropout before conventional clinical signs of DR emerge. Combining AI predictions with diabetes duration data significantly improves diagnostic performance. The identified gene markers further highlight inflammation as a potential driver in early DR, offering novel insights and potential therapeutic targets for preventing DR progression.

## 1. Introduction

Diabetic retinopathy (DR) is a leading cause of vision impairment and blindness worldwide, primarily driven by microvascular complications resulting from prolonged diabetes (type 1 or type 2)-induced hyperglycemia [1,2]. Emerging evidence indicates that glycemic variability also plays a critical role in DR progression [3].

Diabetic retinopathy [4,5,6,7] progresses through two main stages: non-proliferative DR and proliferative DR. Non-proliferative DR represents the early stage of the disease and is characterized by microvascular changes such as the formation of microaneurysms and the narrowing or closure of small blood vessels. At this stage, vision is typically unaffected. However, without timely intervention, non-proliferative DR often progresses to proliferative DR, a more severe stage marked by the growth of abnormal blood vessels in the retina and the formation of fibrous tissue, leading to vision loss. Early diagnosis of DR is therefore critical to prevent disease progression and preserve vision.

Retinal acellular capillaries, or “ghost vessels”, is one of the earliest pathological changes in DR. These non-functional capillaries result in dropout of the basement membrane tubes, disrupting retinal microcirculation [8]. Although these capillary dropouts play a crucial role in the early progression of DR, they remain undetectable by conventional imaging techniques like fluorescein angiography (FA) until more morphological abnormalities, such as microaneurysms, hemorrhages [9], and vascular leakage [10], become apparent. Currently, accurately measuring acellular capillary (“ghost vessels”) density requires destructive techniques such as trypsin digestion to remove non-vascular tissues, making it unsuitable for human patients. The absence of reliable, non-destructive methods to identify diabetes-induced silent retinal capillary dropout presents a significant barrier to early DR detection.

In this study, we aimed to develop a non-destructive approach for the early detection of diabetic retinopathy (DR), prior to the onset of clinically visible morphological abnormalities. To achieve this, we employed the Nile rat (*Arvicanthis niloticus*) as a preclinical model to investigate early microvascular changes associated with DR. The Nile rat is a diurnal rodent native to Northern Africa and represents a highly relevant model for type 2 diabetes and its complications. Unlike conventional laboratory rodents, Nile rats are particularly susceptible to diet-induced diabetes when fed standard laboratory chow, which is hypercaloric relative to their native diet [11]. This metabolic sensitivity mirrors the natural progression of type 2 diabetes in humans. Importantly, diabetic Nile rats develop advanced retinal lesions similar to those observed in human DR, including macular edema, capillary non-perfusion, and proliferative disease [12], making them an ideal preclinical DR model. Using this model, we generated paired datasets consisting of non-destructive fluorescein angiography (FA) images and destructive capillary density measurements obtained via trypsin digestion. We then trained an artificial intelligence (AI) model on the FA images to predict early capillary dropout, defined as an acellular capillary density ≥ 18 counts per mm^2^. Building upon this, we further investigated whether incorporating diabetes duration as prior knowledge within a Bayesian framework could improve model performance. Finally, we conducted retinal transcriptomic profiling to identify gene expression signatures associated with early capillary dropout. Collectively, these efforts aim to address a critical gap in early DR detection by enabling the non-destructive identification of early diabetic retinopathy before clinically visible morphological abnormalities.

## 2. Methods

### 2.1. Animal Protocol and Ethical Statement

All animal experiments were approved by the University of California Santa Barbara (approval code: Protocol #1-24-893.3, approval date: 1 January 2024), Institutional Animal Care and Use Committee, and conducted in accordance with the NIH Guide for the Care and Use of Laboratory Animals; study protocol 893. UCSB founder Nile rats were derived from the Brandeis University colony of the KC Hayes Laboratory. Nile rats in UCSB were housed at 21–26 °C in a conventional facility with individually ventilated cages and were provided autoclaved Sanichips as bedding material. Nile rats were either fed a high-fiber diet (Lab Diet 5L3M; Newco Speciality, Rancho Cucamonga, CA, USA) or a high-caloric diet (Formulab Diet 5008; Newco Speciality, Rancho Cucamonga, CA, USA) [13]. The percentage of crude fiber was 23% for the high-fiber diet and only 4% for the high-caloric diet. The ratio for the percentage of calories provided by carbohydrate, fat, and protein were 67:10:23 for the high-fiber diet and 56:17:27 for the high-caloric diet. Random blood glucose (RBG) levels were measured every four weeks starting at weaning age (4 weeks old). To reduce adverse events from diabetic complications, Nile rats with RBG > 500 mg/dL were euthanized.

### 2.2. Retinal Trypsin Digest and Acellular Capillary Count Reading

From each animal, we did a trypsin digest on retina to count acellular capillary density. The retinal vasculature was loosened from other retinal cells using osmotic lysis, where the retina was placed in RNase-free water for 1 h at 4 °C on a shaker. Next, water was aspirated, and the remaining retinal tissue was incubated in a 10% DNase solution at 37 °C for 15 min. Then, the retina was transferred to a petri-dish with RNase-free water. Working under a dissecting microscope, we used a P200 pipet to squirt water on the retina until the inner limiting membrane was detached and neuronal cells were washed away from the retinal vasculature. For the trypsin digest, we used our previously published method [12]. Micrographs for quantification were taken with a Canon Rebel XSi digital camera (Canon, Tokyo, Japan) attached to an Olympus CKX41 microscope via an LM scope C-mount. For acellular capillary counts, 12 micrographs were taken from 3 randomly selected areas in each of the 4 retinal quadrants and analyzed at a final magnification of 200×. The total area used for quantification corresponded to approximately 8.5% of the whole retinal area. The counts were quantified using FIJI computer software by two masked graders.

### 2.3. Defining Silent (Early and Pre- Pre-Symptomatic) Retinal Capillary Dropout

Acellular capillary density below 10 counts per mm^2^ has been considered very unlikely to be associated with retinopathy [14] and is considered normal. In a prior study [12], the acellular capillary counts per mm^2^ in Nile rats without diabetes reached as high as 16.2, suggesting that densities ranging from 10 to 16 counts per mm^2^ are still less likely to be linked with retinopathy. In this study, based on 124 Nile rats, we observed a significant shift in both random blood glucose levels and diabetes duration when comparing the group with 10 to 16 counts per mm^2^ to the group with 20 to 22 counts per mm^2^ (Appendix A). Specifically, the median random blood glucose increased from 123 mg/dL in the lower density group to 209.4 mg/dL in the higher density group (Wilcoxon test, *p*-value = 0.0221), while the median duration of diabetes increased from 0 weeks to 28 weeks (Wilcoxon test, *p*-value = 0.00801). This indicates a sharp increase in retinopathy risk when the acellular capillary density increases from 16 to 20 counts per mm^2^. Based on this pattern, we define the midpoint (18 counts per mm^2^) as the cutoff for retinal capillary early dropout, marking the point where the likelihood of developing early retinopathy begins to rise substantially.

### 2.4. Random Forest Model to Rank the Feature Importance Which Is Associated with Acellular Capillaries Density

We used a random forest regression model [15] to evaluate the relative importance of multiple predictors for acellular capillary density. Random forest is an ensemble learning algorithm that aggregates the predictions of multiple decision trees, effectively capturing complex, nonlinear relationships among variables. To assess feature importance, we used the mean decrease in accuracy (MDA) method, which quantifies the impact of each feature on model performance. For a given feature X_e_, its importance is measured by the change in prediction accuracy when its values are randomly permuted in the out-of-bag (OOB) samples—data not used in training the corresponding tree. Specifically, the model’s baseline accuracy on the OOB samples, Acc_baseline_t_, is compared to the accuracy after permutation of X_e_, Acc_permuted_t_, for each tree t. The mean decrease in accuracy is then computed as follows:**MDA(X_e_) = (1/T) × Σ_t=1_^T^ (Acc_baseline_t_ − Acc_permuted_t_)**
where T is the total number of trees in the forest. A higher MDA score indicates a greater contribution of the feature to the model’s predictive accuracy, signifying its importance in explaining variation in acellular capillary density.

### 2.5. Fluorescein Angiography (FA) Image-Based AI Model for Retinal Capillary Dropout Detection

We developed an AI model to detect retinal capillary dropout using fluorescein angiography (FA) images. The dataset consisted of 94 images (*n* = 94) from 51 Nile rats, with 57 classified as capillary early dropout (acellular capillary density ≥ 18 counts per mm^2^) and 37 as control (acellular capillary density < 18 counts per mm^2^). FA images underwent preprocessing, including resizing to 224 pixels and center cropping, to ensure consistent input dimensions. All images were normalized using mean values of [0.485, 0.456, 0.406] and standard deviations of [0.229, 0.224, 0.225]. These normalization values are computed from the ImageNet dataset [16] which is a large-scale visual database containing over 14 million annotated images across thousands of object categories and has served as a benchmark for training deep convolutional neural networks. These normalization values are commonly used in the preprocessing pipelines of pretrained models, including DenseNet [17]. The dataset was split into training and validation sets using five-fold cross-validation (K = 5). Each fold used 80% of the data for training and 20% for validation, ensuring that all samples were included in both training and validation across different iterations. Data augmentation was only applied to the training set, incorporating random rotation (10°) and horizontal flipping to enhance model generalization. The validation set was processed without augmentation.

The model was based on DenseNet-169, which consists of four dense blocks and a classifier. Transfer learning was applied by freezing all layers except for the last dense block (DenseBlock4) and the classifier. Fine-tuning was performed to optimize feature extraction for classification. The existing classifier was modified to include a dropout layer (rate = 0.5) for regularization and a fully connected layer with two output neurons for binary classification. The model was trained using the Adam optimizer with cross-entropy loss to measure the discrepancy between predicted and true labels. Hyperparameter optimization was conducted by testing different combinations of epoch numbers, initial learning rates, regularization strengths, batch sizes, and step sizes. The model that achieved the best performance, with an overall accuracy of 80.85%, used the following hyperparameters: epoch number = 30, batch size = 10, initial learning rate = 0.01, and weight decay (regularization strength) = 10^−4^. A step learning rate scheduler reduced the learning rate by a factor of 0.1 every five epochs. During the validation phase, probabilities for each class were computed using the Softmax function (which, for binary classification with two output neurons, is conceptually equivalent to logistic regression) and stored along with the predictions for further analysis. Accuracy for each fold was calculated as the percentage of correct predictions, and the overall accuracy was obtained by averaging the validation accuracies across the five folds to assess the model’s overall performance. The model was implemented using Python (version 3.12.3) and PyTorch (version 2.5.0).

### 2.6. A Bayesian Framework to Integrate the Duration of Diabetes with Real-Time FA-Image AI Model to Predict Capillary Dropout

We developed a Bayesian framework that integrates duration of diabetes with a real-time fluorescein angiography (FA) image-based AI model to improve capillary dropout prediction. First, we used logistic regression to model the relationship between the duration of diabetes (D) and acellular capillary density, estimating the probability of capillary dropout (acellular capillary density greater than 18 counts per mm^2^) as follows:

The prior probability is given by$$P(CD|D)=\frac{e^{\left(\beta_{0}+\beta_{1}D\right)}}{1+e^{\left(\beta_{0}+\beta_{1}D\right)}}$$
where

P(CD|D) is the probability of capillary dropout (CD), defined as acellular capillary density greater than 18 counts per mm^2^) given the duration of diabetes D.β_0_ and β_1_ are regression coefficients.D represents the duration of diabetes.

The maximum likelihood estimation (MLE) is used to find the values of β_0_ and β_1_ of the logistic regression model to maximize the probability of observing the given data. We convert CD into binary outcomes (0 = no CD, 1 = CD). The likelihood function is$$L(\beta_{0},\;\beta_{1})=\prod_{i=1}^{n}P{\left({CD}_{i}\right|D_{i})}^{y_{i}}\;\times \;(1-P{\left({CD}_{i}\right|D_{i}))}^{\left(1-y_{i}\right)}$$

The likelihood function is further log10 transformed as follows:$$\log\;L(\beta_{0},\;\beta_{1})=\prod_{i=1}^{n}[y_{i}logP\left({CD}_{i}\right|D_{i})+\left(1-y_{i}\right)log(1-P\left({CD}_{i}\right|D_{i}))]$$
where$$P(CD_{i}|D_{i})=\frac{e^{\left(\beta_{0}+\beta_{1}D_{i}\right)}}{1+e^{\left(\beta_{0}+\beta_{1}D_{i}\right)}}$$

Given a particular Nile rat, the P (CD|D) which is considered a prior probability will be estimated. We then used a Bayes’ theorem to combine the image-AI-predicted DR probability, denoted as P (CD|AI) with the prior probability P(CD|D)

Then, the AI-image-predicted DR probability, denoted as P(CD|AI), will be integrated with the prior probability P(CD|D) via a Bayes’ theorem to calculate the posterior probability of CD as follows:$$P(CD|AI,D)=\frac{P(CD|AI)\;\times \;P(CD|D)}{P(CD|AI)\;\times \;P(CD\left|D\right)+(1-P(CD|AI))\;\times \;(1-P(CD|D))}$$
where P (CD|AI, D) represents the posterior probability of CD considering both the duration of diabetes and AI-based-imaging-predicted probability.

### 2.7. RNA-Seq Data Analysis

The raw RNA-seq FASTQ files from retina vascular biopsy samples were from our previous study [14] and are available under the GEO accession number GSE220672. We re-analyzed the raw RNA-seq data via mapping to the newly sequenced and annotated Nile rat genome [18]. Specifically, RNA-seq reads were mapped to the Nile rat annotated protein-coding genes using Bowtie [19], allowing up to 2-mismatches. The gene expected read counts and transcripts per million (TPM) were estimated by RSEM [20]. The gene expected counts were further normalized by median-of-ratio method via EBSeq [21] R package. The re-analyzed RNA-seq data via mapping to the Nile rat annotated protein-coding genes was also compared with gene expression values (TPMs and mapping counts) estimated using our previous CRSP (Comparative RNA-seq Pipeline) tool, which does not rely on a sequenced genome. Biomarker detection was conducted based on gene expression values derived from mapping to annotated protein-coding genes in the Nile rat. The EBSeq package [21] was used to assess the probability of gene expression being differentially expressed (biomarkers) between acellular capillary density range from <16 (normal) to 17–18 (capillary pre-dropout) counts per mm^2^. We required that differentially expressed genes should have false discovery rate (FDR) < 5% via and >2 fold-change of median-by-ratio normalized read counts.

## 3. Results

### 3.1. The Association of Duration of Diabetes with Silent Retinal Capillary Dropout

The duration of diabetes is a well-established risk factor for the development of diabetic retinopathy (DR) [22]. However, it remains largely unknown whether the duration of diabetes is directly associated with the onset of silent retinal capillary dropout—a pre-symptomatic feature potentially marking the earliest stage of DR. In this study, we used 124 Nile rats to assess multiple factors which could be potentially associated with silent retinal capillary dropout. We performed trypsin digestion on retinal tissue to quantify acellular capillary density as a measure of silent retinal capillary dropout events. We used a random forest regression model to assess feature importance and rank each factor based on its contribution. The model was trained using diabetes duration, mean random blood glucose (RBG), diabetes status (diabetic vs. non-diabetic), diet (high- vs. low-fiber), age, and sex as input variables, with acellular capillary density as the dependent variable. Feature importance was assessed using mean decrease in accuracy (MDA), which quantifies the increase in prediction error when a given variable is randomly permuted. A higher MDA indicates a stronger contribution of that feature to the model’s predictive power, with larger values reflecting greater importance in determining acellular capillary density. As shown in Figure 1A, diabetes duration, mean RBG, diabetes status, diet, and age are all associated with increased acellular capillary density, while sex has the least impact. Among these factors, diabetes duration emerged as the strongest predictor of acellular capillary density. The correlations between diabetes duration and acellular capillary density are shown in Figure 1B. No significant sex differences were observed when diabetes duration was less than 20 weeks. However, after 20 weeks, males exhibited a slightly greater increase in acellular capillary density compared to females, suggesting that sex may act as a modest conditional modifier dependent on diabetes duration. This suggests that diabetes duration is strongly associated with the early silent retinal capillary dropout which potentially drives the early DR progression.

### 3.2. Fluorescein Angiography Imaging Based Deep Learning Model for Detecting Retinal Capillary Dropout

Although fluorescein angiography (FA) is widely regarded as the clinical gold standard for diagnosing diabetic retinopathy (DR) by detecting hallmark features such as microaneurysms, retinal hemorrhages, and vascular leakage, its potential for identifying retinal capillary dropout, a much earlier sign than these hallmark features observable, remains largely unexplored. A major challenge is that, in human studies, individuals without early DR hallmark clinically defined symptoms, such as microaneurysms, hemorrhages, or leakage—are typically used as controls in training of AI-based imaging models [8,9,10]. This is partly because accurately measuring and quantifying retinal capillary dropout requires trypsin digestion of retinal tissue, a method that is not feasible for human subjects. In this study, we used the Nile rat to generate paired datasets consisting of non- destructive fluorescein angiography (FA) images and destructive capillary density measurements obtained through trypsin digestion. An AI model was then trained on the FA images to predict early capillary dropout, defined as an acellular capillary density of ≥18 counts per mm^2^. We defined a cutoff of 18 counts per mm^2^ which was based on prior findings that densities below 16 counts per mm^2^ are unlikely to be associated with retinopathy [12,14]. Among 124 Nile rats in this study, we observed a marked increase in both random blood glucose levels and diabetes duration in animals with acellular capillary densities of 20–22 counts per mm^2^ compared to those with densities of 10–16 counts per mm^2^. Specifically, the median random blood glucose rose from 123 mg/dL to 209.4 mg/dL (Wilcoxon test, *p* = 0.0221), and the median diabetes duration increased from 0 to 28 weeks (Wilcoxon test, *p* = 0.00801), indicating a substantial elevation in retinopathy risk once acellular capillary density exceeds 16 counts per mm^2^ (Appendix A). Based on this trend, we used 18 counts per mm^2^ as the threshold for defining early capillary dropout, occurring prior to the appearance of typical symptoms such as microaneurysms, retinal hemorrhages, and vascular leakage. Using this cutoff, our AI model identified the FA image as early (“silent”) capillary dropout (acellular capillary density of ≥18 counts per mm^2^) and normal.

The image-AI classification model used Densely Connected Convolutional Networks (DenseNet-169) to predict capillary early dropout versus normal control from FA images. DenseNet-169 is a deep convolutional neural network known for its densely connected layers, where each layer receives feature maps from all preceding layers, enhancing feature propagation and reducing the number of parameters. In this study, we applied transfer learning by leveraging the pre-trained weights of DenseNet-169, which were originally trained on large-scale image datasets, fine-tuning the last convolutional block and the fully connected classification layer to adapt the model for our classification task, while keeping the earlier feature extraction layers unchanged (Figure 2). This approach allowed the model to retain general image feature representations while learning to classify acellular capillary density as exceeding or falling below 18 counts per mm^2^. As shown in Figure 3, the deep learning model can achieve an overall accuracy of 80.85%, with a sensitivity of 84.21%, specificity of 75.68%, and an area under the receiver operating characteristic (ROC) curve (AUC) of 0.86 via a 5-fold cross-validation. Among the total images analyzed, the model correctly identified 48 true positives and 28 true negatives, with 18 incorrect predictions. The robust polynomial fitting curve (Figure 3), generated from correctly predicted cases, suggests that the 15 to 23 counts per mm^2^ range may represent a critical transition window for early DR progression. Within this range, AI-predicted probability aligns strongly with observed trypsin digest-based acellular capillary density counts. This finding suggests that fluctuations in acellular capillary density within this window could be biologically significant for early DR progression. Beyond this range—below 15 or above 23 counts per mm^2^—the AI-predicted probability does not show a strong correlation with observed density values. Specifically, when acellular capillary density exceeds 23 counts per mm^2^ or falls below 15 counts per mm^2^, the AI-predicted probability reaches a plateau, suggesting a potential saturation effect or insufficient sensitivity in these extreme ranges.

### 3.3. Integration of the Duration of Diabetes with AI-Based Real-Time Image Prediction Enhanced the Detection of Early Capillary Dropout

While the duration of diabetes is a well-established and significant risk factor for diabetic retinopathy, not all diabetic patients develop retinopathy, and not all individuals with DR have a long history of diabetes. In humans, one-third of individuals with diabetes will develop DR [2]. More importantly, the timing of DR onset varies among individuals, and the recorded duration of diabetes is often inaccurate in real clinical settings. We investigated whether including the duration of diabetes as a prior knowledge to integrate a real-time image-AI-based Bayesian framework can increase the early capillary dropout detection power. We used a logistic regression to model the relationship between diabetes duration and the probability of acellular early capillary dropout (acellular capillary density ≥ 18 counts per mm^2^). This probability was then incorporated into a Bayesian formula (see Section 2) to compute the posterior probability of real-time AI image-based early capillary dropout detection. As shown in Figure 4A, incorporating the duration of diabetes as a prior with the image AI model resulted in improved classification performance compared to the image AI model alone. The AUC of the ROC curves for the AI-based model alone and Bayesian model incorporating diabetes duration achieves are 0.86 and 0.90, respectively. The difference is statistically significant (DeLong’s test *p*-value = 0.0045). This finding suggests that combining prior knowledge (duration of diabetes) with real-time AI-based image predictions can enhance the reliability and robustness of early capillary dropout detection.

We conducted a simulation study to evaluate all possible combinations of diabetes duration (as prior knowledge) and real-time AI-based image predictions in generating posterior probabilities. As illustrated in Figure 4B, for individuals without diabetes or in the early stages of the disease, the AI image prediction plays a dominant role in detecting capillary dropout. However, for individuals with long-standing diabetes, the likelihood of capillary dropout remains high even when the AI prediction is only moderate. By integrating this Bayesian approach, clinicians can make more informed decisions, especially in borderline cases where AI predictions alone may be uncertain. In practice, this framework could support the prioritization of follow-up screenings, and guide early intervention strategies.

### 3.4. Nile Rat Retinal Vasculature Transcriptome and Gene Markers Associated with Early Silent Retinal Capillary Dropout

Although the Nile rat is a valuable preclinical model for diabetic retinopathy, closely mimicking human disease with progressive retinal vascular lesions such as increased acellular capillaries, pericyte loss, microglial infiltration, vascular leakage, capillary non-perfusion, and neovascularization [12], RNA-seq-based transcriptomic studies remain highly limited. This is because standard RNA-seq workflows require mapping reads to a fully sequenced and annotated genome or transcriptome. However, until recently, no complete genome was available for the Nile rat. To solve this problem, we developed CRSP (comparative RNA-seq pipeline) [23], a tool that integrates multiple comparative cross-species computational strategies, including novel transcript assembly and mapping to annotated mouse transcripts, to impute gene expression levels in the Nile rat. While this comparative species computational approach is valuable, it also introduces potential data noise and technical bias due to differences in genome architectures, sequence divergence, and incomplete cross-species transcript annotation, which may lead to inaccuracies in gene expression quantification. Recently, we released the first fully sequenced and annotated Nile rat genome [18], which could serve as a direct mapping reference. However, like any newly sequenced genome, it still contains unsequenced regions and potential annotation errors. To determine whether our comparative cross-species pipeline or the newly sequenced genome provides more reliable gene expression estimates, in this study we conducted the first benchmarking study to evaluate these two strategies based on our RNA-seq of 28 Nile rat retinal vasculature samples [14].

We first calculated the Spearman’s rank correlation coefficient (Rho) between transcripts per million (TPMs; gene expression values) estimated by CRSP (comparative RNA-seq pipeline), which was used in our prior study, versus TPMs obtained from mapping reads to the newly sequenced and annotated Nile rat genome. The correlation was moderate (Rho = 0.37), suggesting a substantial discrepancy between these two computational approaches. Given that RNA-seq read count distributions are often modeled using a negative binomial (NB) distribution [24,25,26], we next examined whether gene read counts estimated by CRSP and genome mapping both followed the expected NB distribution. In the NB framework, the overdispersion ratio is used to quantify the extent to which variance exceeds the mean. A higher overdispersion ratio indicates greater variability in gene expression, often due to technical noise. In contrast, a lower overdispersion ratio suggests more stable gene expression estimates, with variance closer to the mean, making the data less affected by unwanted variability. To assess this, we calculated the overdispersion ratio for both approaches. We found that transcript read counts estimated by mapping to the newly sequenced Nile rat genome had a statistically significantly lower overdispersion ratio compared to those estimated using CRSP (Figure 5B). This suggests that RNA-seq expression estimates obtained from direct genome mapping exhibit greater reliability than those generated using the CRSP strategy, which relies on cross-species computational imputation and may introduce additional noise due to sequence divergence and incomplete transcript annotation.

To investigate this further, we reasoned that by sorting gene expression values (TPMs) from low to high within each RNA-seq sample and grouping genes across samples based on the same quantile (from 25% to 75% with 5% increments), we could minimize biological variation. This quantile-based approach ensures that variations observed across samples within the same quantile are primarily driven by technical factors rather than biological differences, as genes are ranked solely by expression level. As shown in Figure 5C, when we compared the standard deviations of TPMs within each quantile, gene expression values derived from mapping to the Nile rat genome exhibited significantly lower standard deviations than those estimated using CRSP. This suggests that using the newly sequenced Nile rat genome as a mapping reference yields more consistent gene expression estimates than the CRSP strategy we previously employed.

Therefore, we utilized gene expression data derived from mapping to the newly sequenced Nile rat genome to identify gene markers associated with the early silent retinal capillary dropout.

Since acellular capillary (“ghost vessel”) densities below 16 counts per mm^2^ are considered within the normal range, and densities exceeding 18 counts per mm^2^ are indicative of “capillary silent/early dropout,” we conducted a differentially expressed gene (DEG) analysis comparing retinal vasculature with acellular capillary densities of 17–18 counts per mm^2^—representing a pre-dropout state immediately preceding capillary silent/early dropout—to that of normal retinal vasculature (Figure 6A). We identified 43 DEGs (all upregulated) between normal and pre-dropout retinal vessels (Figure 6B), using a threshold of >2-fold change in normalized read counts and a false discovery rate (FDR) <5%. Interestingly, we observed considerable variability in the expression of these 43 biomarkers among the 24 normal retinal vasculature samples, raising the possibility that baseline molecular differences could be associated with varying degrees of susceptibility to early capillary dropout. Further studies will be needed to determine whether these variations have predictive or functional significance. Notably, among these 43 DEGs, three genes (Bcl2a1, Birc5, and Il20rb) are associated with inflammatory responses. Several studies showed that Bcl2a, Birc5, and Il20rb were associated with chronic inflammation and autoimmune diseases [27,28,29,30], suggesting that inflammation could be one of the factors to drive the early capillary dropout.

## 4. Discussion

Early detection of diabetic retinopathy (DR) remains a significant clinical challenge, especially during the preclinical stage when pathological alterations such as capillary dropout are present but not yet detectable by standard imaging techniques. In this study, we developed an AI-based approach using fluorescein angiography (FA) images to predict early capillary dropout in a Nile rat model of DR. By integrating diabetes duration as prior knowledge within a Bayesian framework, we further enhanced predictive performance. In parallel, transcriptomic study of retinal tissue identified 43 genes associated with the capillary pre-dropout state, including several key pro-inflammatory markers, providing novel molecular insight into early DR pathogenesis.

Our Image-AI model demonstrates the feasibility of non-destructively detecting microvascular abnormal before overt morphological changes occur. This represents a meaningful advance in DR diagnostics, as current imaging techniques, including FA, typically fail to detect early ghost vessels until later stages marked by microaneurysms, hemorrhages, or vascular leakage. By bridging this diagnostic gap, our approach holds potential for earlier clinical intervention, which may delay or prevent progression to vision-threatening stages of DR.

Importantly, incorporating diabetes duration as a prior into the AI model improved predictive performance. This finding underscores the value of integrating longitudinal clinical metadata into machine learning models to capture disease progression dynamics. Our results suggest that leveraging this parameter can enhance model interpretability and diagnostic power.

The transcriptomic analysis further revealed a distinct molecular signature associated with the pre-dropout state. Among the 43 differentially expressed genes, Bcl2a1, Birc5, and Il20rb stood out for their known roles in inflammation and cell survival. Bcl2a1 is an anti-apoptotic gene induced by inflammatory cytokines [31], Birc5 is associated with immune response induced cell survival [29], and Il20rb is a receptor subunit involved in cytokine-mediated inflammatory signaling [32]. The upregulation of these genes suggests that inflammatory processes may contribute to pericyte loss and capillary destabilization prior to morphological vessel damage. These findings support previous work implicating low-grade inflammation as a driver of early DR pathology [33,34,35]. Interestingly, several studies even found an association between diabetic retinopathy and chronic inflammation, autoimmune diseases, such as rheumatoid arthritis [36]. In addition to inflammation-related genes, the early dropout-associated biomarkers encompass diverse biological functions. For instance, *Tal1* promotes sprouting angiogenesis by regulating endothelial tip-cell identity [37]. Lipid transporters such as *Abcb11* and *Pctp* suggest altered lipid trafficking, a stressor for homeostasis [38].

Another unique part of this study is to use the Nile rat as a diabetic retinopathy (DR) model. Compared to other rodent models, Nile rats develop diet-induced type 2 diabetes and manifest DR phenotypes that closely mimic those seen in human patients, including macular edema and capillary non-perfusion. Our ability to pair non-destructive FA imaging with destructive histological assessments in this model enabled robust ground-truth labeling for AI training, an approach that is not feasible in human studies due to ethical and technical limitations.

Despite these strengths, there are several limitations of this study. First, while our AI model performed well in the Nile rat model, its generalizability to human datasets remains to be validated. Second, the use of a single imaging modality (FA) may limit the detection of other subtle vascular changes. Integrating additional modalities such as OCTA (optical coherence tomography angiography) may further enhance predictive power. Lastly, while our transcriptomic data provide valuable insights, future studies incorporating spatial transcriptomics or single-cell RNA-seq could more precisely map gene expression to specific retinal cell types involved in dropout initiation.

Our AI model is specifically designed to detect very-early-stage diabetic retinopathy, prior to the appearance of overt morphological changes such as microaneurysms, hemorrhages, or vascular leakage that are typically visible in imaging like fluorescein angiography (FA). As a result, a key limitation of the model is that it is not trained on data from later stages of retinopathy, which may limit its generalizability across the full disease spectrum. However, it is important to note that later-stage retinopathy is clinically easier to detect using established imaging techniques. Therefore, the model is intended to complement—not replace—existing diagnostic approaches by focusing on the early, subclinical window of disease development when intervention may be most impactful.

In summary, our study demonstrates that early, preclinical retinal capillary dropout can be predicted using AI-enhanced FA imaging, especially when combined with prior knowledge of diabetes duration. The discovery of inflammation-related gene markers further highlights a potentially targetable mechanism in early DR progression.

## Figures and Tables

**Figure 1 biomedicines-13-01926-f001:**
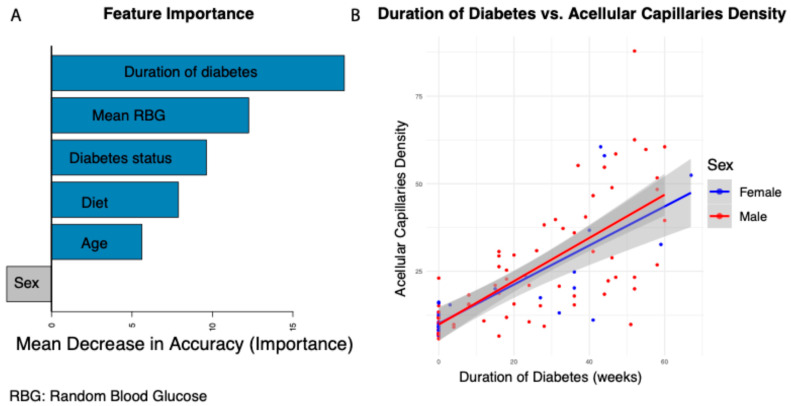
Factors influencing acellular capillary density. (**A**) Feature importance ranking based on the random forest regression model. (**B**) Association between diabetes duration and acellular capillary density.

**Figure 2 biomedicines-13-01926-f002:**
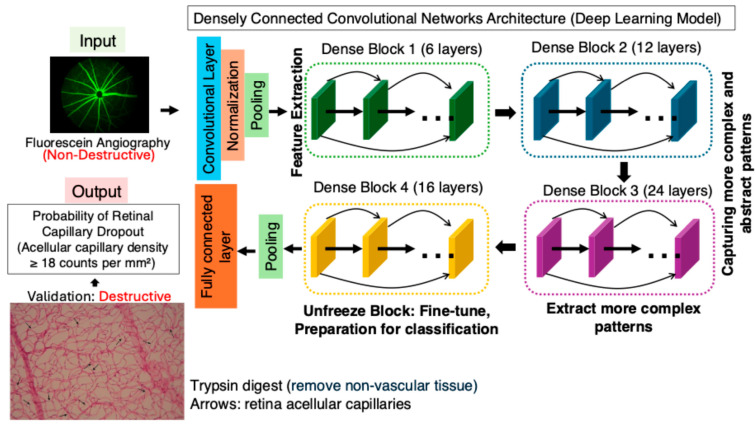
Deep learning model for capillary dropout detection using fluorescein angiography (FA). A DenseNet-169-based AI model was trained on FA images to predict capillary dropout. The ground-truth (acellular capillary density) was based on trypsin digest (destructive) to count acellular capillaries (arrows).

**Figure 3 biomedicines-13-01926-f003:**
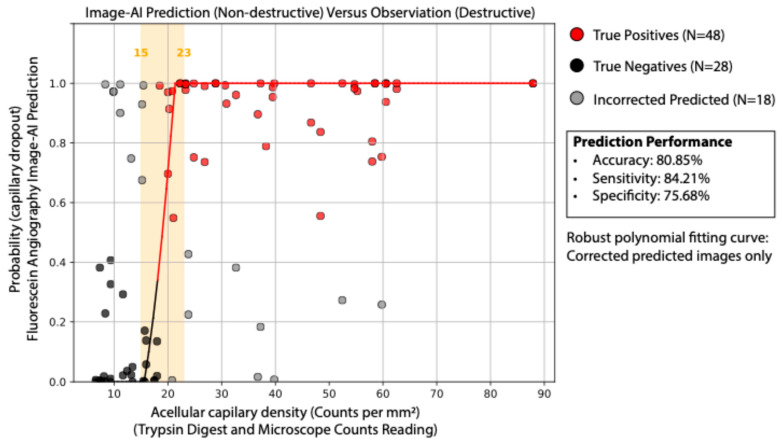
AI-predicted vs. ground truth acellular capillary density. Comparison of FA-based AI predictions with microscopic counts from trypsin digest. The model achieved 80.85% accuracy, 84.21% sensitivity, and 75.68% specificity.

**Figure 4 biomedicines-13-01926-f004:**
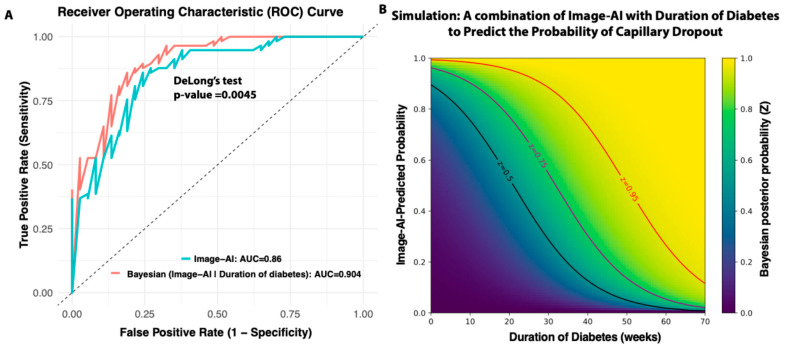
Bayesian framework integrating diabetes duration with AI predictions. (**A**) Improved prediction accuracy using a Bayesian approach. (**B**) Simulation of how diabetes duration combined with AI predictions influences posterior probability of capillary dropout.

**Figure 5 biomedicines-13-01926-f005:**
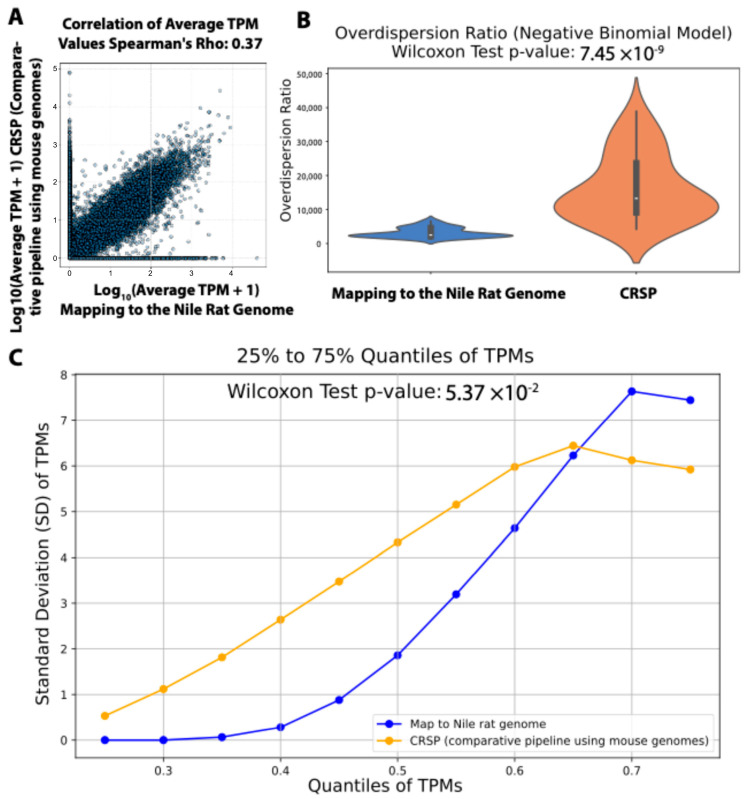
Comparison of gene expression estimates from RNA-seq reads mapping to Nile rat genome vs. CRSP tool (comparative-species pipeline does not rely on a genome). (**A**) Correlation of gene expression values between methods. (**B**) Mapping to the Nile rat genome results in significantly lower over-dispersion than CRSP. (**C**) Comparing the standard deviations of gene expression within the same quantile between these two methods.

**Figure 6 biomedicines-13-01926-f006:**
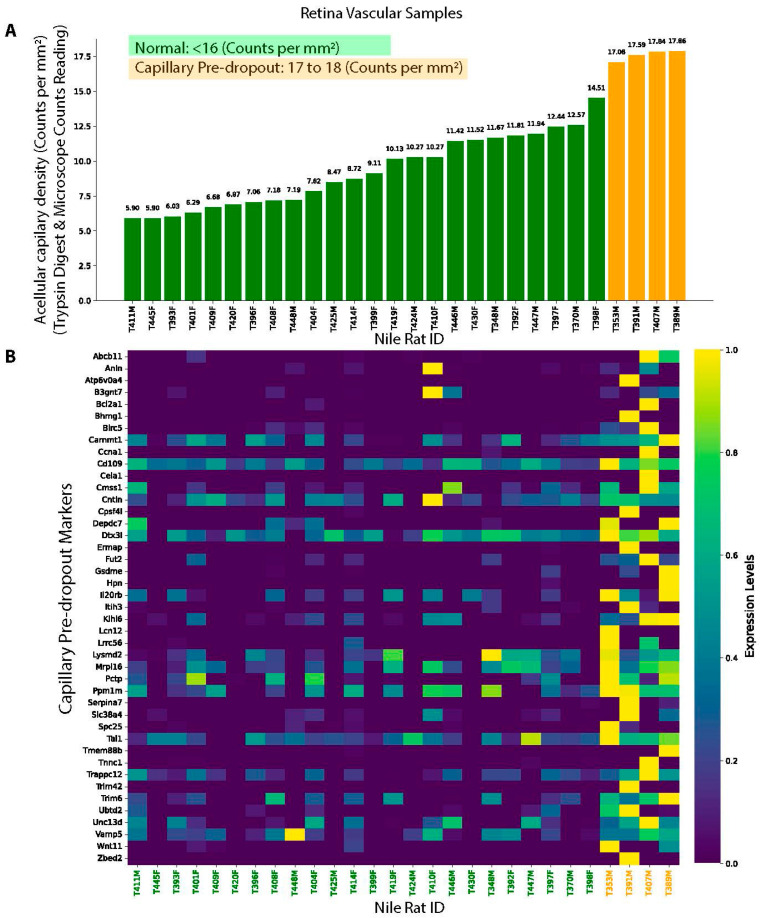
Gene markers associated with capillary pre-dropout state. (**A**) Nile rat RNA-seq samples used for gene marker identification, with green indicating normal and orange representing pre-dropout samples before early DR. (**B**) Identified gene markers associated capillary pre-dropout.

## Data Availability

All data and python codes are available upon request (academic use only).

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
