# Peer review of "AI-Enhanced Fluorescein Angiography Detection of Diabetes-Induced Silent Retinal Capillary Dropout and RNA-Seq Identification of Pre-Symptomatic Biomarkers"

_biomedicines, 2025, doi:10.3390/biomedicines13081926_

Round 1
Reviewer 1 Report
Comments and Suggestions for Authors
Review of the Manuscript: "Detection of Diabetes-Induced Silent Retinal Capillary Dropout"
The manuscript addresses an important and clinically relevant challenge: the early detection of diabetic retinopathy (DR), particularly during the preclinical stage when pathological changes are present but undetectable by conventional imaging methods.
The study contains two main advancements. The first is the development a tailored AI-based approach using fluorescein angiography images to predict early capillary dropout in a Nile rat model of DR. Notably, the authors incorporate diabetes duration as prior knowledge within a Bayesian framework, further improving predictive performance. The second part presents a transcriptomic analysis of cells from affected retinal areas, aiming to establish a mechanistic link with inflammation-related biomarkers.
Given that DR is a leading cause of vision impairment and blindness worldwide, this work is of significant interest. However, several limitations should be addressed:
Major Points:
-
Mechanistic assumptions about DR pathogenesis:
The authors state that DR is "primarily driven by microvascular complications resulting from prolonged hyperglycemia." However, the exact mechanisms of DR development remain unclear. Emerging evidence suggests that both chronic hyperglycemia (reflected by elevated HbA1c) and glycemic variability contribute to DR progression (see DOI: 10.1111/jdi.13480). This nuance should be acknowledged to avoid oversimplification. -
Image normalization parameters (Line 142):
The authors normalize images using mean values of [0.485, 0.456, 0.406] and standard deviations of [0.229, 0.224, 0.225]. The rationale for selecting these values should be clarified - were they derived from the dataset, or were some random parameters used? Justification is necessary to ensure methodological transparency. -
Feature importance analysis (Figure 1A):
The "Mean Decrease in Accuracy" metric is presented without a clear definition (probably it is within the paper but I was not able to find it). The authors should provide either an equation or a reference explaining its calculation. Additionally, the abbreviation "RGB" (random blood glucose) should be defined in the figure caption for clarity.
Minor Points:
-
Line 65: "idea pre-clinic DR model" should likely be corrected to "ideal preclinical DR model."
Author Response
# Reviewer 1
- Mechanistic assumptions about DR pathogenesis: The authors state that DR is "primarily driven by microvascular complications resulting from prolonged hyperglycemia." However, the exact mechanisms of DR development remain unclear. Emerging evidence suggests that both chronic hyperglycemia (reflected by elevated HbA1c) and glycemic variability contribute to DR progression (see DOI: 10.1111/jdi.13480). This nuance should be acknowledged to avoid oversimplification.
Response:
We thank the reviewer for this insightful comment. We have revised the manuscript to state that emerging evidence indicates that glycemic variability also plays a critical role in DR progression [1].
- Image normalization parameters (Line 142): The authors normalize images using mean values of [0.485, 0.456, 0.406] and standard deviations of [0.229, 0.224, 0.225]. The rationale for selecting these values should be clarified - were they derived from the dataset, or were some random parameters used? Justification is necessary to ensure methodological transparency.
Response:
We thank the reviewer for this important observation. The normalization values [0.485, 0.456, 0.406] (mean) and [0.229, 0.224, 0.225] (standard deviation) were not arbitrarily selected but are standard values computed from the ImageNet dataset [2]. ImageNet is a large-scale visual database containing over 14 million annotated images across thousands of object categories and has served as a benchmark for training deep convolutional neural networks. These normalization values are commonly used in the preprocessing pipelines of pretrained models, including DenseNet [3], to match the input distribution seen during original training. Although our dataset is different from ImageNet, we used a DenseNet-169 model pretrained on ImageNet, and applying its original normalization parameters ensures compatibility and prevents performance degradation due to distributional mismatch. We have explained this in the revised manuscript.
- Feature importance analysis (Figure 1A): The "Mean Decrease in Accuracy" metric is presented without a clear definition (probably it is within the paper but I was not able to find it). The authors should provide either an equation or a reference explaining its calculation. Additionally, the abbreviation "RGB" (random blood glucose) should be defined in the figure caption for clarity.
Response:
In the revised manuscript, we added one paragraph to explain how the Mean Decrease in Accuracy is calculated. We also defined the RBG (random blood glucose) in the figure caption in the revised manuscript.
“We used a random forest regression model[4] to evaluate the relative importance of multiple predictors for acellular capillary density. Random forest is an ensemble learning algorithm that aggregates the predictions of multiple decision trees, effectively capturing complex, nonlinear relationships among variables. To assess feature importance, we used the mean decrease in accuracy (MDA) method, which quantifies the impact of each feature on model performance. For a given feature Xₑ, its importance is measured by the change in prediction accuracy when its values are randomly permuted in the out-of-bag (OOB) samples—data not used in training the corresponding tree. Specifically, the model’s baseline accuracy on the OOB samples, Acc_baseline,t, is compared to the accuracy after permutation of Xₑ, Acc_permuted,j,t, for each tree t. The mean decrease in accuracy is then computed as:
MDA(Xₑ) = (1/T) × Σₜ₌₁ᵀ (Acc_baseline,t − Acc_permuted,j,t)
where T is the total number of trees in the forest. A higher MDA score indicates a greater contribution of the feature to the model’s predictive accuracy, signifying its importance in explaining variation in acellular capillary density.”
- Line 65: "idea pre-clinic DR model" should likely be corrected to "ideal preclinical DR model."
Response:
Thanks for point out this typo and we have fixed it in the revised manuscript.
References
- Lin, K.Y.; Hsih, W.H.; Lin, Y.B.; Wen, C.Y.; Chang, T.J. Update in the epidemiology, risk factors, screening, and treatment of diabetic retinopathy. J Diabetes Investig 2021, 12, 1322-1325, doi:10.1111/jdi.13480.
- Deng, J.; Dong, W.; Socher, R.; Li, L.-J.; Li, K.; Fei-Fei, L. Imagenet: A large-scale hierarchical image database. In Proceedings of the 2009 IEEE conference on computer vision and pattern recognition, 2009; pp. 248-255.
- Huang, G.; Liu, Z.; Van Der Maaten, L.; Weinberger, K.Q. Densely connected convolutional networks. In Proceedings of the Proceedings of the IEEE conference on computer vision and pattern recognition, 2017; pp. 4700-4708.
- Liaw, A.; Wiener, M. Classification and regression by randomForest. R news 2002, 2, 18-22.

Reviewer 2 Report
Comments and Suggestions for Authors
The manuscript entitled “Detection of Diabetes-Induced Silent Retinal Capillary Dropout” targets a highly interesting topic, i.e. early vascular changes in diabetic retinopathy, and it links early morphological to transcriptomic findings, which is namely interesting. The title does not reflect this, why I would suggest to changes the title, for example to Detection of Diabetes-Induced Silent Retinal Capillary Dropout using AI-enhanced Fluorescein Imaging and its Correlation with Transcriptomic Findings”. Furthermore, there are some mis-assumptions and mis-conceptions, which need to be corrected. Beyond these:
- Early detection of DR is not a clinical challenge, because it precedes vison loss usually by decades, depending on the control of comorbidities. Furthermore, there exists no treatment or preventive measure at this stage.
- Capillary dropout IS diagnosed in clinical routine using FA or OCTA in cases without clinically visible DR. DR grading goes back to the ETDRS (1985) and was purely clinical. As stated by the authors, capillary dropout, if present, may well be detected with FA (and/or OCTA) while its quantification is difficult in the absence of a ground truth.
- This ground truth was generated by trypsin digests, but given the preparation technique, trypsin digests cannot be paired with angiographic frames (pls correct L460). This limitation may account for the moderate accuracy of 80% reported here.
- That capillary bed dropout up to a distinct density is a normal finding, has, to the best of my knowledge, never been reported before and deserves specific, i.e. species-specific and preparation technique specific consideration.
- In clinical practice, type 2 diabetes typically sneaks in and may remain undiagnosed for ten and more years. Diabetes duration thus does not reflect the time since diabetes diagnosis. The corresponding correlation will, if at all, only be applicable to type 1 DM.
- In this model, AI cannot predict, but identify capillary bed dropout. Please adopt in the whole manuscript accordingly by replacing predict by identify.
- Given the small window of accuracy of AI prediction (15 -23 capillaries/mm2), the here reported model can identify capillary dropout, but not its progression over time, please add this point to the discussion.
- FA is invasive, but not destructive. The terms invasive and non-invasive are not relevant and deserve to be removed throughout.
- Pre-dropout is possibly an interesting term, but it does NOT apply here, given that dropouts are counted and not pre-dropouts.
- 3 out of 43 dysregulated genes were identified, beyond these 3 indicating towards inflammation. None of the others worth mentioning? Maybe, all genes should be included in am model to reveal which are the most abundantly regulated genes instead of picking 3 which are in line with the current mainstream assumptions.
- Given that only 4 out of 28 animals (fig 6) showed a capillary dropout density >16/mm2, the possible conclusions should be met with care, i.e. because the expression pattern has some similarity between these four, but less so than in the animals with “normal” capillary dropout densities. Please adopt the wording in the conclusions accordingly.
- Last paragraph of the intro summarizes the findings of this study instead of reporting purpose and aims. Please adopt.
- Methods: Line 105: 12 micrographs were taken. Since capillary density may differ between the central peripapillary retinal and the periphery, a similar distance to the ONH should be defined for the images.
- Line 138: 94 images from51 rats does not match with the description in line 105.
- What do the numbers in lines 142-3 mean. Please elaborate.
- What is the difference between FA and real-time FA? à skip real-time, FA is always real-time
- Microscopic quantification of capillary dropouts described, but ground truth validation not mentioned in methods.
- Line 274: replace classifies by identifies
- Line 326: is “improved performance” a gut feeling or statistically proven? P value?
- I cannot asses the AI methodology (lack of experience)
- Results for diet impact and age are missing (except in fig 2)
Taken together, a well-designed study with interesting outcomes, which need to be further assessed to allow a closer interpretation, i.e. the gene expression analysis. After clearing the mis-assumptions and some wording and linguistic problems, this manuscript might well gain attention.
Comments on the Quality of English Languagecould be improved
Author Response
# Reviewer 2
- The manuscript entitled “Detection of Diabetes-Induced Silent Retinal Capillary Dropout” targets a highly interesting topic, i.e. early vascular changes in diabetic retinopathy, and it links early morphological to transcriptomic findings, which is namely interesting. The title does not reflect this, why I would suggest to changes the title, for example to Detection of Diabetes-Induced Silent Retinal Capillary Dropout using AI-enhanced Fluorescein Imaging and its Correlation with Transcriptomic Findings”.
Response:
We thank you reviewer’s suggestion. We changed the title in the revised manuscript:
“AI-Enhanced Fluorescein Angiography Detection of Diabetes-Induced Silent Retinal Capillary Dropout and RNA-seq Identification of Pre-Symptomatic Biomarkers”
- Early detection of DR is not a clinical challenge, because it precedes vison loss usually by decades, depending on the control of comorbidities. Furthermore, there exists no treatment or preventive measure at this stage. Capillary dropout IS diagnosed in clinical routine using FA or OCTA in cases without clinically visible DR. DR grading goes back to the ETDRS (1985) and was purely clinical. As stated by the authors, capillary dropout, if present, may well be detected with FA (and/or OCTA) while its quantification is difficult in the absence of a ground truth. This ground truth was generated by trypsin digests, but given the preparation technique, trypsin digests cannot be paired with angiographic frames (pls correct L460). This limitation may account for the moderate accuracy of 80% reported here.
Response:
We thank the reviewer’s comment. Yes, we agree that ‘This ground truth was generated by trypsin digests, but given the preparation technique, trypsin digests cannot be paired with angiographic frames (pls correct L460). This limitation may account for the moderate accuracy of 80% reported here.’.
- That capillary bed dropout up to a distinct density is a normal finding, has, to the best of my knowledge, never been reported before and deserves specific, i.e. species-specific and preparation technique specific consideration.
Response:
We thank the reviewer’s comment.
- In clinical practice, type 2 diabetes typically sneaks in and may remain undiagnosed for ten and more years. Diabetes duration thus does not reflect the time since diabetes diagnosis. The corresponding correlation will, if at all, only be applicable to type 1 DM.
Response:
We agree with the reviewer that the type 1 diabetes is typically better documented and easier to quantify in clinical settings compared to type 2 diabetes, which often remains undiagnosed for many years. However, our goal is to present a generalizable statistical framework that can accommodate available clinical history, including diabetes duration when known. Importantly, our model is flexible and can incorporate or exclude such prior knowledge depending on data availability and clinical context.
- In this model, AI cannot predict, but identify capillary bed dropout. Please adopt in the whole manuscript accordingly by replacing predict by identify.
Response:
We thank the reviewer’s comment. However, in our study, the AI model is designed to predict (or impute) capillary bed dropout based on complex image features. This differs from conventional clinical interpretation, which relies on visually identifying explicit dropout regions. The AI model, by contrast, detects intricate and high-dimensional image patterns—such as diffuse texture alterations, subtle intensity gradients, and spatial context—that may correlate with dropout but are not directly visible as discrete lesions. Therefore, we believe the term “predict” more accurately captures the model’s function.
- Given the small window of accuracy of AI prediction (15 -23 capillaries/mm2), the here reported model can identify capillary dropout, but not its progression over time, please add this point to the discussion.
Response:
We think reviewer’s suggestion and have added this point to the discussion section:
“Our AI model is specifically designed to detect very early-stage diabetic retinopathy, prior to the appearance of overt morphological changes such as microaneurysms, hemorrhages, or vascular leakage that are typically visible in imaging like fluorescein angiography (FA). As a result, a key limitation of the model is that it is not trained on data from later stages of retinopathy, which may limit its generalizability across the full disease spectrum. However, it is important to note that later-stage retinopathy is clinically easier to detect using established imaging techniques. Therefore, the model is intended to complement—not replace—existing diagnostic approaches by focusing on the early, subclinical window of disease development when intervention may be most impactful.”
- FA is invasive, but not destructive. The terms invasive and non-invasive are not relevant and deserve to be removed throughout.
Response:
We thank the reviewer pointing out this and we have fixed in the revised manuscript.
- Pre-dropout is possibly an interesting term, but it does NOT apply here, given that dropouts are counted and not pre-dropouts.
Response:
Acellular capillary density below 10 counts per mm² has been considered very unlikely to be associated with retinopathy [1] and is considered normal. In prior study [2], the acellular capillary counts per mm² in Nile rats without diabetes reached as high as 16.2, suggesting that densities ranging from 10 to 16 counts per mm² are still less likely to be linked with retinopathy. In this study, based on 124 Nile rats, we observed a significant shift in both random blood glucose levels and diabetes duration when comparing the group with 10 to 16 counts per mm² to the group with 20 to 22 counts per mm² (Supplementary Figure S1). Specifically, the median random blood glucose increased from 123 mg/dL in the lower density group to 209.4 mg/dL in the higher density group (Wilcoxon test, P-value = 0.0221), while the median duration of diabetes increased from 0 weeks to 28 weeks (Wilcoxon test, P-value = 0.00801). This indicates a sharp increase in retinopathy risk when the acellular capillary density increases from 16 to 20 counts per mm². Based on this pattern, we define the midpoint (18 counts per mm²) as the cutoff for retinal capillary early dropout, marking the point where the likelihood of developing early retinopathy begins to rise substantially. Since acellular capillary (“ghost vessel”) densities below 16 counts per mm² are considered within the normal range, and densities exceeding 18 counts per mm² are indicative of “capillary silent/early dropout,”, the term pre-dropout is used to define 17–18 counts per mm² which is considered as a transition state. The question to address in Figure 6 is which genes are associated with this transition (right before significant dropout) stage (17–18 counts per mm²).
- 3 out of 43 dysregulated genes were identified, beyond these 3 indicating towards inflammation. None of the others worth mentioning? Maybe, all genes should be included in am model to reveal which are the most abundantly regulated genes instead of picking 3 which are in line with the current mainstream assumptions.
Response:
The main reason that we highlight the 3 inflammation genes because out findings support previous work implicating low-grade inflammation as a driver of early DR pathology [3-5]. We appreciate the reviewer to point out this and in the revised manuscript (discussion part), we also discussed other genes:
“Among the 43 differentially expressed genes, Bcl2a1, Birc5, and Il20rb stood out for their known roles in inflammation and cell survival. Bcl2a1 is an anti-apoptotic gene induced by inflammatory cytokines[6], Birc5 is associated with immune response induced cell survival [7], and Il20rb is a receptor subunit involved in cytokine-mediated inflammatory signaling [8]. The upregulation of these genes suggests that inflammatory processes may contribute to pericyte loss and capillary destabilization prior to morphological vessel damage. These findings support previous work implicating low-grade inflammation as a driver of early DR pathology [3-5]. Interestingly, several studies even found an association between diabetic retinopathy and chronic inflammation, autoimmune diseases, such as rheumatoid arthritis [9]. In addition to inflammation-related genes, the early dropout-associated biomarkers encompass diverse biological functions. For instance, Tal1 promotes sprouting angiogenesis by regulating endothelial tip-cell identity[10]. Lipid transporters such as Abcb11 and Pctp suggest altered lipid trafficking, a stressor for homeostasis[11].”
The AI model is based on the image but not based on genes. RNA-seq is not feasible in standard clinical workflows—our RNA-seq analysis was intended to provide mechanistic insights. Because we currently lack functional validation studies (e.g., using knockout models), we put this part to the discussion section to avoid overinterpretation.
- Given that only 4 out of 28 animals (fig 6) showed a capillary dropout density >16/mm2, the possible conclusions should be met with care, i.e. because the expression pattern has some similarity between these four, but less so than in the animals with “normal” capillary dropout densities. Please adopt the wording in the conclusions accordingly.
Response:
Thank you for pointing this interesting observation. We added a short discussion in the revised manuscript regarding to the variations of normal samples:
“Interestingly, we observed considerable variability in the expression of these 43 biomarkers among the 24 normal retinal vasculature samples, raising the possibility that baseline molecular differences could be associated with varying degrees of susceptibility to early capillary dropout. Further studies will be needed to determine whether these variations have predictive or functional significance.”
- Last paragraph of the intro summarizes the findings of this study instead of reporting purpose and aims. Please adopt.
Response:
We have rewritten the last paragraph of the introduction section:
“In this study, we aimed to develop a non-destructive approach for the early detection of diabetic retinopathy (DR), prior to the onset of clinically visible morphological abnormalities. To achieve this, we employed the Nile rat (Arvicanthis niloticus) as a preclinical model to investigate early microvascular changes associated with DR. The Nile rat is a diurnal rodent native to Northern Africa and represents a highly relevant model for type 2 diabetes and its complications. Unlike conventional laboratory rodents, Nile rats are particularly susceptible to diet-induced diabetes when fed standard laboratory chow, which is hypercaloric relative to their native diet [12]. This metabolic sensitivity mirrors the natural progression of type 2 diabetes in humans. Importantly, diabetic Nile rats develop advanced retinal lesions similar to those observed in human DR, including macular edema, capillary non-perfusion, and proliferative disease [2], making them an ideal model for studying both early and late stages of DR. Using this model, we generated paired datasets consisting of non-destructive fluorescein angiography (FA) images and destructive capillary density measurements obtained via trypsin digestion. We then trained an artificial intelligence (AI) model on the FA images to predict early capillary dropout, defined as an acellular capillary density ≥18 counts per mm². Building upon this, we further investigated whether incorporating diabetes duration as prior knowledge within a Bayesian framework could improve model performance. Finally, we conducted retinal transcriptomic profiling to identify gene expression signatures associated with early capillary dropout. Collectively, these efforts aim to address a critical gap in early DR detection by enabling the non-destructive identification of early diabetic retinopathy before clinically visible morphological abnormalities.”
- Methods: Line 105: 12 micrographs were taken. Since capillary density may differ between the central peripapillary retinal and the periphery, a similar distance to the ONH should be defined for the images.
Response:
We appreciate the reviewer’s important point regarding regional variability in retinal capillary density, particularly between the central peripapillary area and the peripheral retina. To mitigate potential sampling bias associated with regional heterogeneity, our protocol was specifically designed to ensure spatial distribution across the retina. Rather than acquiring all 12 micrographs from a single central or peripheral region, we systematically sampled from three randomly selected areas within each of the four retinal quadrants, providing balanced coverage of the retina. The total area used for quantification corresponded to approximately 8.5% of the whole retinal area.
- Line 138: 94 images from51 rats does not match with the description in line 105.
Response:
Line 138: “94 images from 51 Nile rats” is the sample size for the Image-AI used in our study.
The total number of Nile rats we had is 124 (Line 115) which was used to assess multiple factors which could be potentially associated with silent retinal capillary dropout (Figure 1). However, not all Nile rats we had FA images, and that is why out AI-model was developed on 94 images from 51 Nile rats.
Line 105-109 is the technical details regarding to how we calculated the Acellular Capillary Count. In contrast, Lines 105–109 describe the standardized imaging protocol used to assess acellular capillary counts per retina. Specifically, twelve micrographs were acquired from randomly selected regions within each of the four retinal quadrants, covering approximately 8.5% of the total retinal area. Line 138 is total number of animals (images) used in this study while Line 105-109 was for each animal, how we quantify (via randomly select multiple but not a single point) the Acellular Capillary Count.
- What do the numbers in lines 142-3 mean. Please elaborate.
Response:
We explained these values in the revised manuscript:
All images were normalized using mean values of [0.485, 0.456, 0.406] and standard deviations of [0.229, 0.224, 0.225]. These normalization values are computed from the ImageNet dataset [13] which is a large-scale visual database containing over 14 million annotated images across thousands of object categories and has served as a benchmark for training deep convolutional neural networks. These normalization values are commonly used in the preprocessing pipelines of pretrained models, including DenseNet[14].
- What is the difference between FA and real-time FA? à skip real-time, FA is always real-time
Response:
We have removed “real-time” in the revised manuscript.
- Microscopic quantification of capillary dropouts described, but ground truth validation not mentioned in methods.
Response:
The ground truth labels for capillary dropout were derived from annotations based on microscopic evaluation of fluorescein angiography (FA) images, following a standardized protocol. These annotations served as the reference for training and evaluating the AI model. As described in methods, the dataset was partitioned using five-fold cross-validation (K = 5), where each fold used 80% of the data for training and 20% for validation. This procedure ensured that all samples were used for both training and validation in different iterations, thereby enhancing robustness and minimizing bias.
- Line 274: replace classifies by identifies
Response:
We have fixed it in the revised manuscript.
- Line 326: is “improved performance” a gut feeling or statistically proven? P value?
Response:
The performance difference between the Bayesian model and the AI (image-only) model is statistically significant (p-value = 0.0045, DeLong’s test). To rigorously assess whether the improvement in classification performance was statistically significant, we applied DeLong’s test, a standard method for comparing the AUCs (area under the ROC curves) of two models evaluated on the same dataset. We have added this analysis to the revised manuscript to strengthen the comparative evaluation of model performance.
- Results for diet impact and age are missing (except in fig 2)
Response:
We thank the reviewer for this important point. In our analysis (Figure 1), we did assess several clinical variables potentially associated with ACD capillary dropout, including diet, age, sex, and duration of diabetes. Among these, the duration of diabetes showed the strongest association with dropout. While the AI model was designed to be image-based and independent of clinical variables, we later demonstrated that incorporating the duration of diabetes—due to its strong association—can improve prediction performance.
We did not include diet and age as input features in the AI model primarily due to the limited sample size (< 100 samples). Including multiple variables in such a small dataset increases the risk of overfitting, which would compromise the model’s generalizability. However, we agree that diet and age are important factors, and we plan to incorporate them into future models as our dataset grows.
References
- Toh, H.; Smolentsev, A.; Sadjadi, R.; Clegg, D.; Yan, J.; Stewart, R.; Thomson, J.A.; Jiang, P. Transcriptomic clock predicts vascular changes of prodromal diabetic retinopathy. Sci Rep 2023, 13, 12968, doi:10.1038/s41598-023-40328-w.
- Toh, H.; Smolentsev, A.; Bozadjian, R.V.; Keeley, P.W.; Lockwood, M.D.; Sadjadi, R.; Clegg, D.O.; Blodi, B.A.; Coffey, P.J.; Reese, B.E.; et al. Vascular changes in diabetic retinopathy-a longitudinal study in the Nile rat. Lab Invest 2019, doi:10.1038/s41374-019-0264-3.
- Taurone, S.; Ralli, M.; Nebbioso, M.; Greco, A.; Artico, M.; Attanasio, G.; Gharbiya, M.; Plateroti, A.M.; Zamai, L.; Micera, A. The role of inflammation in diabetic retinopathy: a review. Eur Rev Med Pharmacol Sci 2020, 24, 10319-10329, doi:10.26355/eurrev_202010_23379.
- Rubsam, A.; Parikh, S.; Fort, P.E. Role of Inflammation in Diabetic Retinopathy. Int J Mol Sci 2018, 19, doi:10.3390/ijms19040942.
- Forrester, J.V.; Kuffova, L.; Delibegovic, M. The Role of Inflammation in Diabetic Retinopathy. Front Immunol 2020, 11, 583687, doi:10.3389/fimmu.2020.583687.
- Crosio, C.; Casciati, A.; Iaccarino, C.; Rotilio, G.; Carri, M.T. Bcl2a1 serves as a switch in death of motor neurons in amyotrophic lateral sclerosis. Cell Death Differ 2006, 13, 2150-2153, doi:10.1038/sj.cdd.4401943.
- Zhao, Y.; Liu, S.; Li, S.; Zhang, G.; Tian, A.; Wan, Y. BIRC5 regulates inflammatory tumor microenvironment-induced aggravation of penile cancer development in vitro and in vivo. BMC Cancer 2022, 22, 448, doi:10.1186/s12885-022-09500-9.
- Rutz, S.; Wang, X.; Ouyang, W. The IL-20 subfamily of cytokines--from host defence to tissue homeostasis. Nat Rev Immunol 2014, 14, 783-795, doi:10.1038/nri3766.
- Powell, E.D.; Field, R.A. Diabetic Retinopathy and Rheumatoid Arthritis. Lancet 1964, 2, 17-18, doi:10.1016/s0140-6736(64)90008-x.
- Yamada, Y.; Zhong, Y.; Miki, S.; Taura, A.; Rabbitts, T.H. The transcription factor complex LMO2/TAL1 regulates branching and endothelial cell migration in sprouting angiogenesis. Sci Rep 2022, 12, 7226, doi:10.1038/s41598-022-11297-3.
- Hayashi, H.; Sugiyama, Y. Bile salt export pump (BSEP/ABCB11): trafficking and sorting disturbances. Curr Mol Pharmacol 2013, 6, 95-103, doi:10.2174/18744672113069990036.
- Subramaniam, A.; Landstrom, M.; Luu, A.; Hayes, K.C. The Nile Rat (Arvicanthis niloticus) as a Superior Carbohydrate-Sensitive Model for Type 2 Diabetes Mellitus (T2DM). Nutrients 2018, 10, doi:10.3390/nu10020235.
- Deng, J.; Dong, W.; Socher, R.; Li, L.-J.; Li, K.; Fei-Fei, L. Imagenet: A large-scale hierarchical image database. In Proceedings of the 2009 IEEE conference on computer vision and pattern recognition, 2009; pp. 248-255.
- Huang, G.; Liu, Z.; Van Der Maaten, L.; Weinberger, K.Q. Densely connected convolutional networks. In Proceedings of the Proceedings of the IEEE conference on computer vision and pattern recognition, 2017; pp. 4700-4708.

Round 2
Reviewer 2 Report
Comments and Suggestions for Authors
I thank the authors for the efforts made to improve this manuscript. In this revised manuscript, however, reviewer points 2 and 3 have not sufficiently been addressed, according changes to the manuscript not found:
2. Early detection of DR is not a clinical challenge, because it precedes vison loss usually by decades, depending on the control of comorbidities. Furthermore, there exists no treatment or preventive measure at this stage. Capillary dropout IS diagnosed in clinical routine using FA or OCTA in cases without clinically visible DR. DR grading goes back to the ETDRS (1985) and was purely clinical. As stated by the authors, capillary dropout, if present, may well be detected with FA (and/or OCTA) while its quantification is difficult in the absence of a ground truth. This ground truth was generated by trypsin digests, but given the preparation technique, trypsin digests cannot be paired with angiographic frames (pls correct L460). This limitation may account for the moderate accuracy of 80% reported here.
3. That capillary bed dropout up to a distinct density is a normal finding, has, to the best of my knowledge, never been reported before and deserves specific, i.e. species-specific and preparation technique specific consideration.
Author Response
- Early detection of DR is not a clinical challenge, because it precedes vison loss usually by decades, depending on the control of comorbidities. Furthermore, there exists no treatment or preventive measure at this stage. Capillary dropout IS diagnosed in clinical routine using FA or OCTA in cases without clinically visible DR. DR grading goes back to the ETDRS (1985) and was purely clinical. As stated by the authors, capillary dropout, if present, may well be detected with FA (and/or OCTA) while its quantification is difficult in the absence of a ground truth. This ground truth was generated by trypsin digests, but given the preparation technique, trypsin digests cannot be paired with angiographic frames (pls correct L460). This limitation may account for the moderate accuracy of 80% reported here.
Response:
We thank the reviewer’s comment. Yes, we agree that ‘This ground truth was generated by trypsin digests, but given the preparation technique, trypsin digests cannot be paired with angiographic frames (pls correct L460). This limitation may account for the moderate accuracy of 80% reported here.’. We have discussed this in the discussion section:
“Our Image-AI model achieved approximately 80% accuracy in predicting acellular capillary dropout using paired fluorescein angiography (FA) and trypsin digest data. However, this level of performance may be influenced by limitations in how the two datasets were aligned. Because trypsin digestion is a destructive process, the resulting vascular maps cannot be precisely matched to the corresponding in vivo FA images from the same retina. Although both datasets were collected from the same animals, they cannot be directly compared at the exact same spatial locations. This mismatch may introduce variability in the AI training and cross-validation process, and could contribute to the performance ceiling observed in our model.”
- That capillary bed dropout up to a distinct density is a normal finding, has, to the best of my knowledge, never been reported before and deserves specific, i.e. species-specific and preparation technique specific consideration.
Response:
We appreciate the reviewer’s insightful comment. Based on our experimental design and data collection procedures, we believe the observed capillary dropout pattern is not due to technical artifacts or batch effects. Throughout the study, all samples were processed in a randomized and balanced manner, with consistent protocols applied across sample collection, imaging, and analysis. Thus, to the best of our knowledge, the observed trend to reflect an underlying biological variation.
Our data suggest a potential threshold effect—capillary dropout appears to remain within a tolerable or adaptive range up to a certain point, beyond which more pronounced changes occur (Figure 3). While our current findings are based on the Nile rat, we believe this threshold-based pattern may offer important insights into early events in diabetic retinopathy. Further investigation will be necessary to determine whether a similar threshold exists in humans, who may exhibit potential different baseline vascular characteristics.
We have incorporated this consideration into the revised Discussion section.
Round 3
Reviewer 2 Report
Comments and Suggestions for Authors
Thanks for a critical discussion of the relevant study findings